

# Should oasification be ignored when examining desertification in Northwest China?

Dongwei Gui[1,2*], Jie Xue[1,2,3,4], Yi Liu[1,2,3,4], Jiaqiang Lei[1,2], Fanjiang Zeng[1,2]

[1] State Key Laboratory of Desert and Oasis Ecology, Xinjiang Institute of Ecology and Geography, Chinese Academy of Sciences, Urumqi 830011, Xinjiang, China

5    [2] Cele National Station of Observation and Research for Desert-Grassland Ecosystems, Cele 848300, Xinjiang, China

[3] Key Laboratory of Biogeography and Bioresource in Arid Zone, Chinese Academy of Sciences, Urumqi 830011, Xinjiang, China

[4] University of Chinese Academy of Sciences, Beijing 100049, China

*Corresponding to*: Dongwei Gui (guidwei@ms.xjb.ac.cn )Tel: +86-0991-7885469; Fax: +86-0991-7885320

**Abstract.** There has been substantial expansion of oases in the Northwest of China in recent
decades; however, research has largely focused on desertification rather than oasification and its
associated mechanisms. Based on the description of desertification in arid regions and other
regions, and analysis of oasification in arid or hyper-arid areas, we firstly elucidate the
theoretical relationship between oasification and desertification. We then examine the current
understanding in oasification research, and propose that this will enhance our understanding of
desertification through the integration of studies of oasification. Finally, we suggest three
potential topics for future oasification research. We also propose a simple conceptual model for
the identification of an appropriate oasis size under different land uses, including the crucial
assessment of associated uncertainties. Based on the analysis of oasification research from an
epistemological and methodological perspective, we suggest that oasification should have equal
research status to desertification in arid regions.

## 1 Introduction

Oases are unique intra-zonal landscapes located within deserts in arid regionsthat allow
vegetation and human settlements to flourishunder improved water availability [*Zhang et al.,*
*2003a; Nasierding and Zhang,* 2009]. They are formed in river deltas, alluvial plains, the edges
of diluvial fans, or under the increased runoff and groundwater from nearby mountains [*Liu et al.,*
*2010; Song and Zhang,* 2015]. Usually oases are classified into natural oases and artificial oases
according to their formation mechanism. Natural oases include desert riparian forests, valley
meadows, and shrubs. Artificial oases develop from desert or natural oases and include
agricultural oases, city or town oases, and industrial oases [*Han and Meng,1999; Guo et al.,*



*2016*]. More than 30% of land worldwide [*Okin et al.,* 2006] and 22% of the land area in Northwest China [*Bai et al.,* 2014] is classified as arid. Therefore, oases are located within the arid regions areas of China, which are mainly distributed between the west of Helan Mountain and the north of the Qinghai–Tibetan Plateau (Figure 1). In this region, oases are the basis of

human life and economic development, supporting more than 95% of the population and more than 90% of social wealth with only around 5–6% of the land surface [*Han and Meng,*1999*; Jia et al.,* 2004]. In Northwest China the stability of the oasis ecosystem directly relates to regional sustainable development and therefore the maintenance and stabilization of the oasis ecosystem is an important research topic.

However, research has become increasingly complex and difficult under expansion of oasis area because of the stresses of rapid population growth and socioeconomic development. Oasis expansion (oasification) is the process of conversion of desert to oasis under the activities of humans and human–environment interactions [*Wang,* 2009*; Gui et al.,* 2011]. Oasification and desertification are basic geographic processes in arid areas [*Han and Meng,*1999*; Gui et al.,*

2010]; however, oasification receives little attention compared with desertification. Based on the search result from ISI Web of Science between 1996 and 2016, we found that there are around 4000 papers using "desertification" as the key word but less than 40 papers mentioning "oasification". Scholars from China have conducted more research into oasis evolution than those of other nations [*Ishiyama et al.,* 2007]. Even so, the ratio of oasification to desertification

research still is less than 1:100 according to the Chinese Science Citation Database. These results indicate that previous research into geographic processes or oasis evolution in arid areas has focused on desertification and oasification has been largely ignored.

In addition, a considerable body of research has emphasized that oasification and desertification are two opposite geographic processes in arid regions [*Zhang et al.,* 2003*; Su et al.,* 2007*; Xie et al.,* 2014*; Song and Zhang,* 2015], and it is rarely suggested that they are two complementary processes in the course of oases evolution [*Wang and Li,* 2012]. Therefore, the

relationship between oasification and desertification in arid area remains to be determined. It also remains unclear whether it is appropriate to research oasis stability or its evolution from the perspective of desertification only, without considering oasification. If oasification should not be ignored, the focuses of oasification research need to be defined. In accordance with these questions, this research examines oasis evolution characteristics and the current state of the

research progress in China. The aim of this paper is to (1) clarify the relationship between oasification and desertification in arid regions; (2) discuss the importance of oasification research; and (3) suggest important topics for future oasification research.

## 2 The relationship between desertification and oasification in arid regions

### 2.1 The description of the desertification processes

The concept of desertification defined by the International Convention to Combat Desertification[*ICCD,* 1994] has been widely accepted. It explicitly describes desertification as a process that occurs in arid, semi-arid and dry sub-humid regions. Oasification was first proposed in an international journal by *Mart ńez* [2000] and to date, there remains no unified definition of the phenomenon. In China, scholars widely describe oasification as mostly occurring in arid or

hyper-arid areas with annual precipitation less than 200mm, known as the oasis distribution area [*Luo et al.,* 2006]. Notably, the area of desertification includes the area of oasification. To improve understanding of the desertification process in different areas, we produced a schematic



description of desertification in arid areas (precipitation lower than 200 mm) and other areas (precipitation higher than 200 mm) as Fig. 2.

Figure 2 shows that the causes of desertification are completely different between the different areas. In the semi-arid and dry semi-humid regions, the landscape is covered by relatively abundant vegetation, which can be seen as "steady state", such as shrubs and trees, with annual precipitation of about 200–600 mm [*Abahussain et al.,* 2002; *Cai et al.,* 2004; *Wang et al.,* 2016].However, mainly due to human overgrazing or other inappropriate land-use activities, the desert gradually replaces the "steady state" vegetation, resulting in species decrease and land degradation. Thus, the desert is an "invasive object" and desertification is an "invasive process" in these areas. Nevertheless, when appropriate management practices are implemented, such as limiting stocking density, the desertification can be controlled and the original "steady state" can recover gradually under the existing climate background.

In arid or hyper-arid areas, the situation is different. The desert is already in a "steady state" with less vegetation and less than 200mm annual precipitation under the climate background [*Gui et al.,* 2016].However, owing to snowmelt runoff or springs derived from mountain areas flowing into desert, oases appear as a non-zonal landscape. These oases support local population and ecosystem services. However, the oases expand at different spatial and temporal scales under population and economic pressures. Thus, the oases can be seen as "invasive objects" and oasification as an "invasive process". The threat of desertification remains until the dynamic balance between oases and desert is reached, otherwise desertification will recapture its territory under appropriate climatic conditions, especially when the water resources cannot support oasis expansion.



2.2 The relationship between desertification and oasification in arid areas

Oasification is usually classified as the opposite of desertification (shown in Fig. 2b),because oasification and desertification produce different landscapes and their results and processes are opposite.

However, this opposite relationship is not the sole characteristic between these processes. The causes of desertification presented in Fig. 2b indicate that unsuitable oasis expansion results in desertification in arid or hyper-arid areas. Therefore, in this situation oasification and desertification become integrated and this "unity" becomes a characteristic of the two processes. The complete relationship between oasification and desertification is the "unity of opposite" and

thus ignoring any parts of this relationship is unscientific when considering geographic processes in arid regions.

**3 The importance of oasification research**

Based on the research into changes to oases and the oasification situation in China in recent decades, we next list the reasons that oasification research should not be ignored.

3.1 Enriching oasis science and enhancing desertification research in arid regions

The understanding of the oasis evolution process could provide corresponding theories for oasis exploitation [*Xie et al.,* 2014; *Xie et al.,* 2015]. Currently, research related to oasis evolution has mainly focused on four aspects: analysis ofland use patterns and their driving factors in oasis regions [*Guo et al.,* 2008; *Bai et al.,* 2014; *Wang et al.,*2014; *Song and Zhang,*

2015], discussion ofthe relationship between oasis evolutionand water resources [*Siebert et al.,* 2007; *Liu et al.,* 2010], assessment of the ecological effects of oasis evolution[*Su et al.,* 2007; *Zhang et al.,* 2014], and determination  on the suitable oasis scale [*Ling et al.,* 2013; *Guo et al.,*

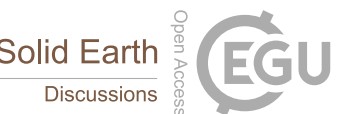

2016]. These studies mainly concentrated on agricultural oases since these represent the typical landscape surrounding oases. However, this research has focused mainly on detailed analysis or assessment of specific cases, and has ignored the basic theoretical study of oasis evolution. Previous research into oasis evolution can all be classified into the "oasification" category.

Under this "oasification", the research logic of oasis evolution will be more clear than the present situation. Similarly, the land degradation research logic became gradually clear after the "desertification" was established. Thus, emphasizing oasification study is beneficialnot only for integrating oasis evolution but also for enriching oasis science more widely.

Moreover, the oasification research can also be considered a supplement and enhancement of

the desertification research in arid or hyper-arid regions. Desertification, as it occurs in the semi-arid and dry semi-humid regions, is an invasion process. Although oasification can also be seen as an invasion process in arid areas, it is more suitable to consider it as a restoration process (Fig. 2). Therefore, just as desertification research mainly focuses on the causes or driving forces in other areas, we also should pay attention to oasification when discussing desertification in arid

areas because the oasification could be a contributing factor. At present, a considerable amount of research has emphasized that desertification is the major obstacle to sustainable development of oases in Northwest China [*Zhang et al.,* 2003a; *Su et al.,* 2007; *Nasierding and Zhang,* 2009;*Li et al.,* 2015], and seldom mentioned that oasification itself may represent an obstacle. We could therefore achieve a more realistic analysis of desertification if we proceed from the

perspective of oasification. After all, during thousands of years of oasis evolution, the threats of desertification have consistently occurred and the existence and disappearance of oases are largely related to the abundance or shortage of water resources [*Lu et al.,* 2003; *Jia et al.,* 2004; *Zhang et al.,* 2014]. Thus, unsuitable oasis expansion alongside the inappropriate utilization of



water resources will finally result in desertification. Correspondingly, strengthening oasification research is important in our understanding of the mechanisms of desertification and could help us to better control desertification in arid regions.

3.2 The expansion of oases in China has continued throughout recent decades

Oases have expanded in Northwest China since the People's Republic of China was founded in 1949 [*Han and Meng,* 1999; *Zhao et al.,* 2008; *Gui et al.,* 2011; *Zhou et al.,* 2016; *Zhao et al.,* 2016]. For example, the area of oases in Xinjiang province (the area in which the oases are mainly distributed in China) has doubled since 1949 based on remote sensing image analysis [*Gui et al.,* 2011]. Oasification during this period can be divided into three stages. The first stage was before China's Reform and Opening (1980s) and was mainly driven by population growth and policy. The second stage was from 1980 to 2000 and the driving factors were mainly economic improvements. Although the oasis area substantially increased during these two stages, the greatest degree of expansion has occurred in the third stage (from 2000 to the present), which was mainly driven by water utilization technologies. For example, since 2000, the drilling well cost has decreased remarkably and groundwater utilization has become easier. In addition, advanced irrigation techniques such as drip irrigation have been popularly adopted, which has directly turned a large area of desert into oasis (Fig. 3).

The recent oasis expansion includes two characteristic processes. The first is that the cultivated land was mainly changed to an agricultural landscape such as cropland or orchard with higher water consumption. This not only changes the spatial distribution pattern of water resources in the river basin, but also results in overexploitation of local groundwater. The agricultural activities usually consume much more water in the upper reaches of the river and inevitably influence water supply, riverbeds, and land in the lower reaches. Moreover, they also

destroy the local ecosystem and could cause fixed and semi-fixed sand dunes to become active

again [*Zhang et al.,* 2003b; *Moulin et al.,* 2006]. The Tarim River and the Heihe River (the

longest and second longest inland river, respectively) are typical examples of this situation [*Xie

et al.,* 2014; *Jiang et al.,* 2015]. The second characteristic is these oases have mainly expanded

into natural oases, especially within the desert-oasis ecotone, which representsan important

ecological protective screen for artificial oases. A reduction in the size of natural oases or the

change from woodland and grassland to arable land will result in a more fragile environment and

increase the likelihood of land desertification [*Piao et al.,* 2005; *Nasierding and Zhang,* 2009].

Although advanced irrigation techniques have already been gradually popularized in oases, they

can contribute to problems such as the rapid expansion of soil salinization, the decline of the

groundwater, and vegetation degradation in the interior of the oasis. Those problems are

ubiquitous in the arid areas of China [*Zhang et al.,* 2003a; *Su et al.,* 2007; *Wang and Li,* 2012].

They prompt us to urgently consider which type of oasis expansion is suitable and how to control

or govern our activities and protect natural systems in these arid areas of China. Therefore,

studies of oasification have important practical value.

## 4 Important topics in oasification research

Drawing on insights from the inherent characteristics of oasis evolution, we next consider the

topics that should be the focus of future oasification research.

4.1 Research into the dynamic balance between oases and desert in the mountain–oasis–desert

(MOD) system

There can be no doubt that water plays the key role in the creation, development, and

extinction of oases. Arrival of fresh groundwater or surface runoff from thawed glaciers, snow,

and orographic rainfall in nearby mountains profoundly influences the location and the size of an oasis. Therefore, studies of oasis evolution should begin from water resources under the specific MOD system. An oasis can be classified into three statuses within the MOD system: expanding, shrinking and dynamic balance. Regardless of whether an oasis is initially expanding or

shrinking under the change of water resources, it will eventually reach a dynamic balance with the surrounding desert. Hence, understanding the mechanisms governing the dynamic balance between oasis and desert should be a key research topic. During the oasis–desert interaction, the mechanisms should be primarily analyzed from the perspective of matter, energy, and information flows [*Ma et al.,* 2015; *Li et al.,* 2016]. Furthermore, it requires an in-depth study of

hydrological processes, soil water transport processes, and evapotranspiration processes. Historically, the change of oasis status was mainly influenced by climate. For instance, the oasis status in a warm period is entirely different from that in a cool period [*Zhang et al.,* 2003a; *Li and Chao,* 2015]. However, the range of temperature fluctuation at a hundred-year scale is much smaller and is therefore unlikely to cause substantial variation of the water resources in oases.

Human activities, therefore, are the major current influence of variations of oasis status through direct or indirect influences of water resources [*Zhang et al.,* 2003a; *Nasierding and Zhang,* 2009]. Therefore, the question of how to combine natural and human factors to reveal the dynamic change mechanism of oases is particularly important.

4.2 Choice of oasis size and assessment of associated uncertainties

An oasis has an optimal area, which varies for  different land-use types, and allows a dynamic steady state to be achieved based on water resource replenishment rate and use level [*Han and Meng,* 1999; *Liu et al.,* 2010]. In China, an oasis is usually surrounded by agricultural land-use,which accounts for the largest area and is the most important part [*Wahap et al.,* 2004;*Cheng*



*et al.,* 2006;*Gui et al.,* 2010]; shelterbelt and natural vegetation located in oasis–desert ecotones, which play an important role in sheltering farmland against damage from sandstorms and strong winds [*Metzger et al.,* 2006; *Wang et al.,* 2007]; and city or town, for human residence and industrial development. The structure of the oasis system determines its function, and the sum

area of different land-use types determines the oasis size. The size of an oasis is not necessarily "the larger the better". When the size of an oasis exceeds the carrying capacity of the water resource, the oasis stability will be challenged, and could potentially lean the region towards desertification. Study of the appropriate oasis size based on its specific location in watershed and its function is always an important topic in oasis research [*Han and Meng,* 1999; *Ling et al.,*

2013; *Gui et al.,* 2016].Climatic factors (determining the volume of surface water and groundwater) and human factors (determining oasis structure) both have associated uncertainties and therefore estimates of appropriate oasis size must include uncertainty quantification. Therefore, study of the appropriate oasis size must be conducted, particularly focusing on the quantification of uncertainties.

Based on the water balance principle of an oasis, we provide a conceptual model for suitable oasis size and discuss the uncertainties of the variables and parameters. For an oasis size denoted as *A*, the oasis water balance *f(A)* can be expressed as follows:

$$f(A) = W_R + W_G - W_B - W_L - ET \tag{1}$$

where $W_R$ is the runoff into the oasis; $W_G$ is the groundwater recharge to the oasis; $W_B$ is the deep

seepage; $W_L$ is the water consumption e.g., for residential and industrial uses; and *ET* is the oasis evapotranspiration from agricultural land, shelterbelts and nature vegetation. This can be further expressed as follows:

$$ET = \gamma \times A \times ET_1 + (1-\gamma) \times A \times ET_2 \tag{2}$$



where $ET_1$ is the unit area ET of farmland, $ET_2$ is the unit area ET of ecological land such as shelterbelt and natural vegetation, and $\gamma$ is the ratio of farmland in oasis. In China, the $\gamma$ usually is higher than 70% based on remote sensing image analysis. Of the ET components, $ET_1$ has a close relationship with water use efficiency ($WUE$) and the agricultural yield ($Y$) [*Gui et al.,*

5  *2016*].This relationship can be expressed as follows:

$$WUE = \frac{Y}{ET_1} \tag{3}$$

indicating that when the yield ($Y$) remains approximately constant, the $ET_1$is negatively correlated to $WUE$. This means that changes to irrigation schemes (such as flooding, the installation of drip irrigation, or changes to its irrigation period) will influence $ET_1$. Through

10  replacing $ET$ in Equation (1) with Equations(2) and (3), $f(A)$can be expressed as follows:

$$f(A) = W_R + W_G - W_B - W_L - [\gamma \times A \times \frac{Y}{WUE} + (1-\gamma) \times A \times ET_2] \tag{4}$$

If we know every variable and parameter listed on the right side of Equation (4), we can approximate the water balance situation for the current oasis size, i.e. $f(A)$. When $f(A)$is less than zero, the oasis is overexploiting water resources and the oasis is subject to an unsuitable

15  expansion process. Continuation of this expansion would lead to severe ecological and environmental problems. In an ideal situation, if we define the suitable oasis size is $A^{'}$, then $f(A^{'})$ should be equal to zero, and the equation can be expressed as follows:

$$0 = W_R + W_G - W_B - W_L - [\gamma \times A^{'} \times \frac{Y}{WUE} + (1-\gamma) \times A^{'} \times ET_2] \tag{5}$$

After equation transformation, $A^{'}$ can be calculated as follows:

$$A' = \frac{W_R + W_G - W_B - W_L}{\gamma \times Y / WUE + (1-\gamma) \times ET_2} \tag{6}$$

Based on Equation (6), we can state that both natural variables and human variables or parameter have uncertainty attributes that will finally propagate to uncertainties in the oasis size estimation. Moreover, Equation (6) can be expanded to be more complex in terms of the included parameters and contain more uncertainty information. For example, even in the

agricultural land-use types, different crops have different evapotranspiration levels. The traditional flooding irrigation is also gradually being replaced by drip irrigation, which will strongly influence the oasis ET. In addition, under the urbanization policy of China, the land-use type is also undergoing changes; for example, in the south rim of Tarim Basin, the traditional farmlands such as cotton have transformed to more economically attractive fruit forests. This

will influence oasis structure and further influence the suitable oasis size through the corresponding changes to water use and allocation. These factors can be added to Equation (6) as variables or parameters. Thus, determination of the appropriate oasis size for an area should not only couple natural and human factors but also consider the uncertainties associated with these factors. As a result, the optimal oasis size should be shown as a range rather than a definite

number. Unfortunately, present studies have mostly provided a constant value to suggest an appropriate oasis size, typically calculated based on the water volume of each valley and the oasis water demand [*Ling et al.,* 2013; *Gui et al.,* 2016]. To our knowledge, no studies have considered the uncertainty associated with all the included factors.

## 5 Conclusions

Oases are essential components of arid ecosystems that are extremely fragile and sensitive to anthropogenic disturbance under scarce precipitation and water resources [*Xie et al.,* 2015;*Xue et al.,* 2016]. Oasis science, an important subject branch of arid area science, has recently focused on the following major issues: changes to oases under land use changes such as irrigated



agriculture, the appropriate use and management of water, oasis bearing capacity, soil and land use, and biology and climate in oasis-desert ecotones [*Li and Chao,* 2015]. Many such research topics should have been explored under the topic of "oasification."

Owing to superimposed influence of natural and social dynamic factors, the oasis system

isalways in an active and dynamic state, and it is necessary to observe and study oases from a dynamic and development perspective[*David et al.,* 2001; *Jordán et al,* 2003;*Wang et al.,* 2007]. However, most research has discussed these dynamic characteristics from the perspective of desertification only, perhaps because the concept of desertification is more popular in international academia. Correspondingly, basic oasification theoretical research has mainly been

ignored. Similar to oases and desert, oasification and desertification are independent concepts that interact with each other [*Li and Zhou,* 2016] and thus these concepts should be consistently considered as independent and yet interconnected. After our description of desertification in arid regions and other regions, we suggest that the completed relationship between oasification and desertification in arid or hyper-arid area is a "unity of opposite" from the epistemology angle.

This approach to discussing oasification will not lead to confusion or contradiction with desertification. On the contrary, it will help us to better understand oasis evolution. After all, oasification is an invasion process and desertification is a restoration process in the arid or haper-arid areas. In the context of decades of oasis expansion in China, research into this topic has become practically important. The establishment of studies will not only improve understanding

and modelling of desertification but will also integrate and enrich oasis science.

Water is a vital limiting factor for the sustainable ecological, economic, and social development of an oasis. Hence, oasification research must focus on water as the main variable during under the watershed scale in MOD system. Three key topics should be considered in

oasification research. First, the dynamic balance mechanism between oasis and desert must be elucidated to identify the balance point. Second, natural and social or human factors should be coupled completely, which is also an international research frontier in water resource management [*Ayvaz and El ġi,* 2013]. Third, studies of the optimal oasis size and associated

uncertainties within specific watersheds are necessary to maximize economic and ecological benefits. This will also contribute to the optimization of water and land-use research. One of the major challenges in ecological modeling of water and land-use is the assessment of their uncertainties[*Fu and Guillaume,* 2014; *Vrugt et al.,* 2008; *Lu et al.,* 2014], so uncertainty analysis should be practically emphasized.

Geographic processes in arid regions have become an active research field involving climate system, global change, land degradation, and regional sustainable development [*Cheng et al.,* 2006; *Xie et al.,* 2015]. As one of the vital research highlights of global land use and cover change, oasis change has been regarded as an important content of regional environmental change research [*Xie et al.,* 2014]. In China, as a country with wide arid area and oasis

distribution, the geographic processes should be discussed not only from the perspective of desertification but also from the perspective of oasification. Research efforts in this area will remain incomplete should we continue to emphasize one and minimize or ignore the other.

**Acknowledgments**

This work was financially supported by the National Natural Science Foundation of China

[U1603343,41471031].



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



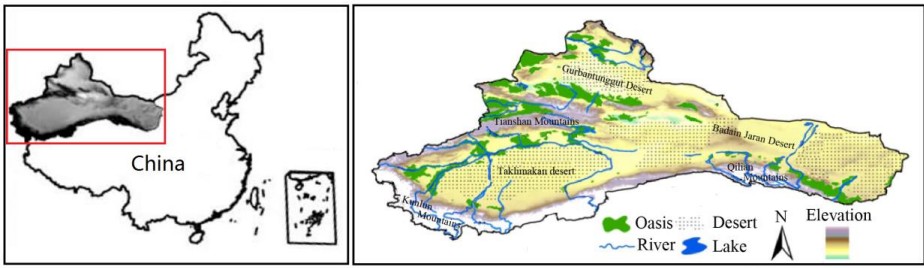

**Fig. 1 Distribution of oases in Northwest China**






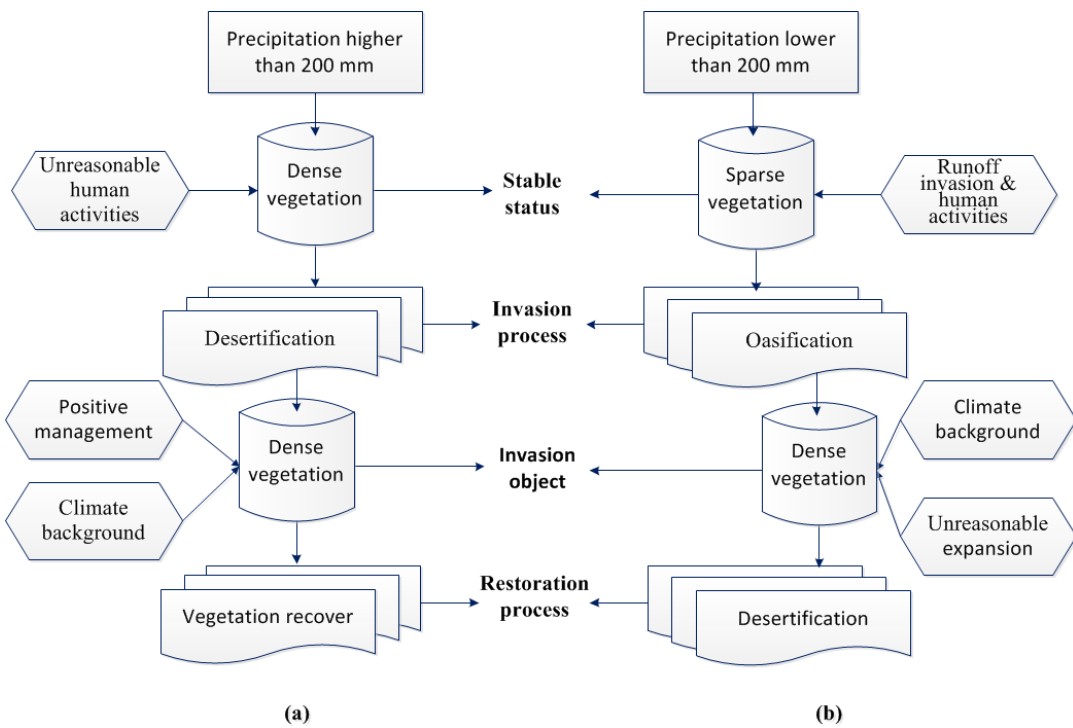

**Fig.2 Schematic diagramto describe the desertification and oasification processes in semi-arid and dry semi-humid regions (a), and in arid or hyper-arid regions (b)**

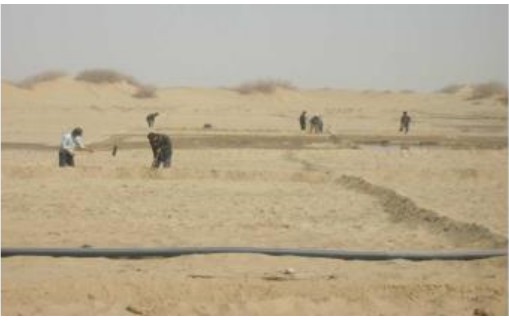

**Fig. 3 Reclamation ofsanddunes and transformation into agricultural land in the Tarim Basin**



