# Peer review of "Should oasisification be ignored when examining desertification in Northwest China?"

_Solid Earth, 2017_

## Referee Comment (RC1) · Anonymous Referee #1 · 31 Aug 2017

SE-2017-59 Manuscript Title: Should oasification be ignored when examining desertification in Northwest China? Authors: Dongwei Gui et al.

Comments:

The subject is very interesting, hence it deserves to be proposed in the SE journal. The basic scientific question in the manuscript was: how the relation between oasification and desertification processes happen (Page 3 Line 13-14). The manuscript draws attention to the process of oasis expansion, which its subject was clearly treated as the major issue for the research (Page 3 line 15). Other objectives were a) to discuss about the importance of oasi-desert relation research and b) to propose topics for future researches. However, in the manuscript there are certain gaps. I listed below some major weakness points which need to be addressed: a) In fact, what are the

major mechanisms involved in oasis-desert relation? They should be presented accurately and, also observe the good balance of importance among the mechanisms in their presentation. The manuscript did not establish a strong connection between the mechanisms of the oasification-desertification processes. I think that this can be the major lack in the manuscript. b) In the subsection 2.1. (Description of the desertification process) the readers expect information about the vegetation, water and soil qualities found in oasis and deserts. Concerning the aspects of the desertification process, is the information clear to readers? c) For the readers, it does not clear the anthropogenic pressure contribution in the oasification or desertification process (subsection 2.2 and conclusions). d) The figures are not clear and they have low information quality provided in the text. The descriptions of the figures in the manuscript should be improved to provide a better understanding of the context by readers (mainly figures 2 and 3). e) The subsection 4.2 (Choice of oasis size. . .) was presented in a detailed way while the 4.1 subsection did not. In fact, the human activity role in oasification process was underachieved (4.1 subsection). f) On the line 5, page 10, the phrase is too general to explain the process accurately. g) Are you sure that the subtitle 2.2 should repeat the title 2? h) Why the manuscript was proposed as a Short communication instead of a review article? Dear editor, according to your request I have completed my review of the manuscript. In my opinion, the manuscript could not be published at the present state. The Reviewer

---

## Short Comment (SC1) · 1 Sep 2017

We appreciate reviewer's comment. Here we answer reviewer questions as follows. Actually, in our manuscript, we want express a point: why the oasification shouldn't be ignored in NW China. After review the research situation on the oasification, we set three layers or three parts in logic in order to answer this question:firstly we compare the background of desertification and oasification happening; secondly, we introduced the importance of oasification research; lastly, we provided the research direction of oasification. These three parts consitute the whole manuscript. Therefore, this manuscript is more of an opinion paper than a review paper, this is why we select it as Short communication type. a) The reviewer think the basic scientific question in the manuscript was "how the relation between oasification and desertification processes happen", it is

partly right, because of it only appeared in our first part or layer (subsection 2), and we already used the logic map (fig.2) to explain the process of desertification and oaisfication in different area, as to the mechanisms of the oasification-desertification processes including more detail information and we think it can be find in many papers and unsuitable repeat again. b) In subsection 2.1 reviewer think we should provide more detail information about soil, vegatation and so on in the desertification process, we think in this comment or opinion paper, it is unsuitalbe to list such case study or detail infomation. But we will present more detail research information in the future research. c) In subsection 2.2 and conclusion, reviewer think " it does not clear the anthropogenic pressure contribution in the oasification or desertification process", actually we express it in the subsection 3 (the second layer/part),we listed the three stages (page 8, line 9). d) Reviewer think figure 2 and figure 3 has low information. Actually, the figure 2 is the essence and the most important part in this manuscript, it reflects the logic of desertification and oaisfication process. And fig.3 is just show readers how the situation of oasification in NW China. Since we set the paper is comment or opinion type, so we just provide more simple and logic map rather than specific data map. e) The subsection 4 is our third layer or third part, it just simple introduced how to research oasification in the future, 4.2 is a concept model so we provide more detail information. Reviewer hope we can provide more detail information like this, but just we mentioned as above, as one short comments paper, we only select important part to list detail information. f & g) we will carefully think about how to explain accurately.

---

## Referee Comment (RC2) · Anonymous Referee #2 · 5 Sep 2017

Dear Editor,

1. the abstract need major revision. the objectives are too broad. the author need to focus on one of the points as stated in the manuscript rather than focus on-all-three aspect. build up the idea of the equation and focus to one specific point. a review paper is better suited to summarize if the author choose to focus on 3 aspect at once. 2. in the model equation, many variables put in place are highly open for discussion, thus not conclusive. 3. introduction is acceptable, with minor revision. 4. results and discussion are too vague. difficult to understand the focus of the oasification study as suggested by the author. 5. figures are not supportive of the discussion/idea of improvement for oasification research. 6. the flow chart is still at initial stage of development, thus need further clarification on the flow. 5. the conclusion too long and not focused.

[Figure]

unfortunately, the manuscript is rejected.

---

## Short Comment (SC2) · 7 Sep 2017

Much appreciate reviewer's comment. 1. The reviewer think our paper "The objectives are too broad", actually, we did this on prupose. We want to emphersize the importance of oasification research, we also want to call for more scientists to focus on oasification. Thus we hope this paper more broad rather than narrow, that is, it is an opinion paper not an case study. Our purpose is to let more scientists understand oasification and its importance. 2. In the model equation, actually it is a conception model and just tell readers one direction, there are still lots work to do, and it is also difficult to provide a model can be used in everewhere. 3. For land degradation research scientists, especially focus on land degradation in arid area, we think the idea in this paper is easy understand, oasification is one easy conception. 4. All figures in this paper

eigher support our opinion or introduce the situation of oasis/oasification.

The reviewer think the main problem in this paper is "too broad" or "too vague", actually, as an opinion paper, a logic paper, an conception paper, we believe many scientist don't understand if they don't fimilar this research field, however we also believe many scientist will understand its logical, especially figure 2. We think research logic is very important, begining from oasification to research land degradation is more suitalbe than begining from desertification, this our core logic. We hope editor can support our opinion.

---

## Author Comment (AC1) · 18 Oct 2017

Dear editor and referees, Much appreciate for your valuable advices to our manuscript on behalf all authors. After carefully reading your comment, we revised and even reorganized our manuscript based on referees' advices, and any change can be found in the track change version. We hope you can further give us valuable suggestions. As a scientist focusing on oasis research about 20 years, I know the problem we meet in the oasis research field. we think oasis science need development, and not only for oasis itself but also for better understanding desertification in arid area. This manuscript is opinion or discussion or conception type, we only hope provide our viewpoint to international readers and let more scientist to discuss or criticize it, and finally to make it perfect. Thank you very much! Dr. Dongwei Gui On behalf of all co-authors

[Figure]

We list our response as follow one by one. RC1: 1. The subject is very interesting, hence it deserves to be proposed in the SE journal. A: Thank you for your encourage. 2. The basic scientific question in the manuscript was: how the relation between oasification and desertification processes happen (Page 3 Line 13-14). A: Actually, as a short communication, this manuscript mainly want to express why the oasification shouldn't be ignored in NW China. The relationship between oasification and desertification is only one part to answer our question and has been clear in the part 2, especially in figure 2. 3. The manuscript draws attention to the process of oasis expansion, which its subject was clearly treated as the major issue for the research (Page 3 line 15). A: Yes, because this topic is ignored more or less in the international research, and even in China. 4. Other objectives were a) to discuss about the importance of oasis-desert relation research and b) to propose topics for future researches. A: Firstly, "to discuss about the importance of oasis-desert relation research" is not our objective. The oasis and oasification are different concepts, just like desert and desertification. We only discuss the importance of oasification and its relation with desertification in arid area. Then we propose topics for future researches. In the first version, it is expressed not very clear, and we have re organized in the revised version. 5. However, in the manuscript there are certain gaps. I listed below some major weakness points which need to be addressed: a) In fact, what are the major mechanisms involved in oasis-desert relation? They should be presented accurately and, also observe the good balance of importance among the mechanisms in their presentation. A: There are many researches to discuss the "mechanisms involved in oasis-desert relation". In this revised version, we have simply introduced the basic situation in page 2, and in the page 14 line 9-22. This manuscript is submitted as short communication type, we want pay more attention to present the importance of oasification research on oasis sustainable development and on desertification research, then we hope to arouse more researchers' attention on oasification rather than just desertification in arid area. As to the " mechanisms involved in oasis-desert relation " research can be found in many literatures. 6. The
manuscript did not establish a strong connection between the mechanisms of the oasification-desertification processes. I think that this can be the major lack in the manuscript. b) In the subsection 2.1. (Description of the desertification process) the readers expect information about the vegetation, water and soil qualities found in oasis and deserts. Concerning the aspects of the desertification process, is the information clear to readers? A: In the revised version, the section 2 was used to explain: a) the desertification process occurred in semi-arid and dry semi-humid regions (figure 3a); b) desertification and oasification process in arid and hyper-arid regions (figure 3b). Actually, the figure 3 is the essence and the most important content in this manuscript. The figure 3 is a conception flow chart, it help us to understand the logical process of oasification and desertification happen in arid area. As to " readers expect information about the vegetation, water and soil qualities found in oasis and deserts ", we think it is not the topic in the manuscript and theses detail information can be found in many researches. We more hope answer the title question from a macro style. 7. c) For the readers, it does not clear the anthropogenic pressure contribution in the oasification or desertification process (subsection 2.2 and conclusions). A: In the revised version, we use one section to introduce "the anthropogenic pressure contribution in the oasification or desertification process " , it is in page 5 line 9-22, and page 6, page 7 line 1-10. 8. d) The figures are not clear and they have low information quality provided in the text. The descriptions of the figures in the manuscript should be improved to provide a better understanding of the context by readers (mainly figures 2 and 3). A: In the revised version, we have modified the figure 3 (it is figure 2 in the revised version) to clear show the process of transforming desert into oasis. However, the original figure 2 ( it is figure 3 in the revised version) is conception map, we didn't change. Since we set the paper is comment or opinion type, we just provide more simple and logic map rather than specific data map. 9. e) The subsection 4.2 (Choice of oasis size. . .) was presented in a detailed way while the 4.1 subsection did not. In fact, the human activity role in oasification process was underachieved (4.1 subsection). A: This subsection (it is subsection 3.2 in the revised version) is simple introduced how

to research oasification in the future. Actually, the style of this section is just like other parts, which actually is also a simple style, it is only a concept model. However, as one modes with equations, we need show reader how this model is derived. 10. f) On the line 5, page 10, the phrase is too general to explain the process accurately. A: The phrase is "Regardless of whether an oasis is initially expanding or shrinking under the change of water resources, it will eventually reach a dynamic balance with the surrounding desert". We have changed as "the oasis scale will eventually reach a dynamic balance with the surrounding desert under the water resource limit". g) Are you sure that the subtitle 2.2 should repeat the title 2? A: We have reorganized this section. In the revised version, We have change subsection title of 2.1 and 2.2 as " The process of oasification and desertification in arid areas ", " Oasification research is supplement of desertification research in arid area ", respectively. 11. h) Why the manuscript was proposed as a Short communication instead of a review article?. A: Since we want express our opinion that the oasificaiton shouldn't be ignored, we think Short communication is enough. After all it is not case study. RC2: 1. the abstract need major revision. the objectives are too broad. the author need to focus on one of the points as stated in the manuscript rather than focus on-all-three aspect. build up the idea of the equation and focus to one specific point. a review paper is better suited to summarize if the author choose to focus on 3 aspect at once. A: In the revised version, we have provided a major revision for abstract. It is correct that the objectives are two broad in the original version. In the revised version, we have narrow our paper purpose, in the page 4 line 23 to page 5 line 8, we mentioned "The aim of this research is to elucidate the importance of oasification and its research through 1) examining oasification characteristics in recent decades in Northwest China, and 2) clarifying the logical relationship between oasification and desertification in arid regions. Then, based on the current state of the oasis research progress, we propose important topics for future oasification research. Finally, we hope to arouse more researchers' attention on oasification rather than just on desertification in arid area." Since the purpose of this manuscript is to answer why the oasification shouldn't be

ignored in NW China. In order to answer this question clearly, we need elucidate the oasification characteristics in recent decades in Northwest China. We need to understand the logical relation between oasification and desertification in arid area in theory. However, it is not enough to just emphasize the importance of oasification research. We need to propose its research direction for reader to discuss. We think these contents are together to answer our question, and it is not completed if ignore anyone aspect. 2. in the model equation, many variables put in place are highly open for discussion, thus not conclusive. A: Yes, we set this paper as viewpoint/discussion paper, and just want express our understanding in oasification, and we prepare each section as more macro perspective rather than micro perspective or case study. In the model equation, actually it is a conception model. We established the model and provide parameters to readers, it is also discussed as a macro perspective. If we use a case study to use the model, the paper will become one another type, we will do that in the future. 3. introduction is acceptable, with minor revision. A: Yes, we have revised and any change can be found in the new verison. 4. results and discussion are too vague. difficult to understand the focus of the oasification study as suggested by the author. A: We have tried our best to revised, and change can be found in the track-changes version. 5. figures are not supportive of the discussion/idea of improvement for oasification research. A: There are three figures in this manuscript. In the revised version, figure 1 shown readers the oases distribution in NW China, it is one basic figure. Figure 2 (it is figure 3 in the original version) is just shown readers how the oasification process happen in NW China, this figure is not clear, we have revised. Figure 3 (it is figure 2 in the original version) is a conception map, help us to understand the logical process of oasification and desertification happen in arid area . We think the figure 3 is the essence and the most important content in this manuscript, if we provide a detail flow chart, it is difficult to reflect these logical relation between oasification and desertification in arid area. 6. the flow chart is still at initial stage of development, thus need further clarification on the flow. A: Just we mentioned on above. As one viewpoint paper, we hope express the relation between oasification and

desertification happen with a logic way. 7. the conclusion too long and not focused. A: In the revised version, the "Conclusion" section has been changed as "Discussion and conclusion" . And, we have revised this section and try to focused.

Please also note the supplement to this comment:
https://www.solid-earth-discuss.net/se-2017-59/se-2017-59-AC1-supplement.pdf

―――――――――――――――